# Relationship among state reopening policies, health outcomes and economic recovery through first wave of the COVID-19 pandemic in the U.S.

**Alexandre K. Ligo**[1,2], **Emerson Mahoney**[1], **Jeffrey Cegan**[1], **Benjamin D. Trump**[1], **Andrew S. Jin**[1], **Maksim Kitsak**[3], **Jesse Keenan**[4], **Igor Linkov**[1,5]*

**1** US Army Corps of Engineers, Engineer Research and Development Center, Concord, Massachusetts, United States of America, **2** Department of Engineering Systems and Environment, University of Virginia, Charlottesville, Virginia, United States of America, **3** Faculty of Electrical Engineering, Mathematics and Computer Science, Delft University of Technology, Delft, The Netherlands, **4** School of Architecture, Tulane University, New Orleans, Louisiana, United States of America, **5** Department of Engineering and Public Policy, Carnegie Mellon University, Pittsburgh, Pennsylvania, United States of America

☯ These authors contributed equally to this work.

* igor.linkov@usace.army.mil

**Data Availability Statement:** The data underlying the results presented in the study are available from the New York Times and the Opportunity

## Abstract

State governments in the U.S. have been facing difficult decisions involving tradeoffs between economic and health-related outcomes during the COVID-19 pandemic. Despite evidence of the effectiveness of government-mandated restrictions mitigating the spread of contagion, these orders are stigmatized due to undesirable economic consequences. This tradeoff resulted in state governments employing mandates at widely different ways. We compare the different policies states implemented during periods of restriction ("lockdown") and reopening with indicators of COVID-19 spread and consumer card spending at each state during the first "wave" of the pandemic in the U.S. between March and August 2020. We find that while some states enacted reopening decisions when the incidence rate of COVID-19 was minimal or sustained in its relative decline, other states relaxed socioeconomic restrictions near their highest incidence and prevalence rates experienced so far. Nevertheless, all states experienced similar trends in consumer card spending recovery, which was strongly correlated with reopening policies following the lockdowns and relatively independent from COVID-19 incidence rates at the time. Our findings suggest that consumer card spending patterns can be attributed to government mandates rather than COVID-19 incidence in the states. We estimate the recovery in states that reopened in late April was more than the recovery in states that did not reopen in the same period– 15% for consumer card spending and 18% for spending by high income households. This result highlights the important role of state policies in minimizing health impacts while promoting economic recovery and helps planning effective interventions in subsequent waves and immunization efforts.

Insights Team at https://github.com/
OpportunityInsights/EconomicTracker.

**Funding:** I.L., J.C. and B.T., were funded in part by
the US Army Engineer Research and Development
Center FLEX-4 Project on Systemic Resilience. The
funders had no role in study design, data collection
and analysis, decision to publish, or preparation of
the manuscript.

**Competing interests:** The authors have declared
that no competing interests exist.

## 1. Introduction

The COVID-19 disease pandemic caused by the SARS-CoV-2 virus prompted U.S. states to
develop policies restricting population mobility and economic activity, with vastly different
timelines and degrees of intensity [1, 2]. Even with considerable uncertainty in the early
months of the pandemic regarding viral pathogenicity and downstream health outcomes, U.S.
state policy (inspired by the Guidelines to Reopen America Again [3]) sought to limit socio-
economic activities with varying degrees of risk as vectors of viral transmission—with hopes
of gradual easing of policy restrictions as (a) genomic and pandemic surveillance improved
through more rigorous testing, and (b) the incidence and prevalence rates declined to a level
suggesting community spread was a minimal risk [4]. The intent of such policy measures was
to limit any economic disruption to brief periods of time (e.g., 15 to 30 days). What occurred,
however, was a divergence in the duration and intensity of socioeconomic and mobility
restrictions, as well as controversy regarding the socioeconomic and public health conse-
quences that such restrictions (colloquially, "lockdowns") may have such as reduction in
spending and increase in COVID-19 incidence over time.

This paper examines how economic indicators such as consumer card spending vary based
upon the timing of state decisions to enter and exit socioeconomic and mobility lockdowns,
and how the economic trends and lockdown policy measures differ based on SARS-CoV-2
incidence rates. Our analysis sheds light on how much the temporal differences in incidence
rates, lockdowns and reopening decisions across states are associated with their economic
recovery and ultimately resilience, as defined as the ability to absorb and recover from disrup-
tions [2]. This paper examines the first wave of responses to the COVID-19 pandemic in the
U.S. between mid-March and early August of 2020 and discusses deviations from expected
outcomes in several U.S. states. We find that state decisions to reopen following the first wave
of economic lockdowns seem to have a sizeable effect on economic recovery. We estimate that
in states that reopened between the 20th and 27th of April, recovery in consumer card spending
was 15.2% more than the recovery in states that did not reopen in the same period. The esti-
mated effect of in consumer card spending among higher income households is higher. For
the same period, we found the recovery in the states that reopened was 18.2% more than in the
states that did not reopen. Therefore, while there is strong evidence among existing literature
and in our findings that state regulations are not the sole cause for all the economic and behav-
ioral changes that persist during the pandemic, these regulations do have a significant effect.

## 2. Background and contribution of this paper

Studies show that COVID-related lockdowns have a variety of different effects (both qualita-
tively and quantitatively) in public health and the economy [5, 6], and the impact of restric-
tions on COVID-19 incidence, prevalence rates and economic outcomes might not be
straightforward. Fowler et al. show that counties that implemented lockdown orders saw a
decline in new COVID-19 cases by 30 percent after just one week [1]. Furthermore, Abouk
and Heydari [7] show a 37 percent decrease in new cases fifteen days after a county imple-
mented lockdown measures. Amuedo-Dorantes et al. [8] estimate that implementing the lock-
down dates by one day earlier would have reduced nationwide COVID-19 death rates by 2.4
percent.

Others have examined the early economic consequences of SARS-CoV-2 in the United
States and globally, and report controversial findings. For example, Baker et al. found that
spending increases early in the pandemic possibly due to stockpiling, followed by sharp
declines in spending in late March 2020 as cases began to spread [9]. In epidemic models of
consumer behavior, people tend to cut back on both consumption and labor hours during

epidemics in order to reduce their chances of getting infected [10, 11]. In addition, Guerrieri et al. argue that even when supply shocks are concentrated to specific industries they can have economy wide repercussions, including in sectors that were not impacted by the supply shocks [12]. Lin and Meissner [6] show that states did not have statistically different employment outcomes depending on whether or not "lockdown" orders were implemented. Chetty et al. [13] discussed whether consumer spending has recovered as a result of the economic stimulus implemented through the CARES Act [14], and noted that the stimulus alone could not account for spending increases of the magnitude observed. Although the stimulus payments could have played an important role in creating more optimistic consumer mindsets and rebounding consumer spending levels, other factors should explain the totality of consumer spending recovery. In particular, Chetty et al. found a relative difference in recovery for five states that reopened (which we discuss in detail in this paper). Goolsbee and Syverson [15] found that only seven points of the 60 percent decline in consumer traffic came from lockdown orders, suggesting that individual choices and fear of infection were more important than legal restrictions. In contrast, Coibion et al. [16] deployed a survey-based study to analyze how consumers would react to lockdowns in terms of spending habits and macroeconomic expectations in April 2020. They found that respondents in lockdown-afflicted households expected the unemployment rate to be 13 percent higher over the following twelve months and higher unemployment rates for three to five more years as compared to households that did not go into lockdowns. They also expect low inflation, higher uncertainty, low mortgage rates, and they moved out of foreign stocks to more liquid forms of investment. While the effect of restrictive policies on economic collapse and subsequent recovery was often measured at the industry level [6] and at the county level, aggregated by U.S. counties of different incomes [13], it is quantitatively unclear how the effect of those policies on economic recovery varied with individual state interventions.

While the aforementioned literature focuses on relevant aspects of COVID-19 spread, public policy, and economic impact, the contribution of this paper is on addressing the effect of these three dimensions on each other. We analyze restrictions and reopening non-pharmaceutical interventions (NPI) and how those decisions were related both with COVID-19 spread and with economic indicators in the states examined. The analysis targets the impact of the lockdown on consumer card spending in each state, and the rate of spending recovery once state restrictions were lifted. The main research question that we address is how critical indicators of economic activity vary based upon state decisions to enter and exit lockdowns and trends in COVID-19 incidence. This is important because reopening decisions implemented too early may have resulted in increased spread of COVID-19, while states that reopened too late may have caused unnecessary economic hardship. Furthermore, we discuss whether economic disruption and recovery is affected by state decisions or COVID-19 incidence.

## 3. Materials and methods

We have gathered time series data at the state level from publicly available sources to examine the relationship between state interventions, COVID-19 spread and economic impact during early restrictions and reopening interventions in the U.S. For this purpose, we examine the variation of COVID-19 incidence and consumer card spending over time. The data, sources and operations performed to integrate and analyze the data are described below.

### 3.1. COVID-19 data

For the analysis in this paper, we have used new COVID-19 confirmed infections per day (incidence) per 100,000 people in each state as an indicator of the disease spread. We have

chosen incidence as a metric because of its overall availability in publicly available datasets such as the New York Times and other sources. The choice of incidence is consistent with the CDC recommendation for states to track the trajectories of COVID-19 infections along with other metrics. We considered that indicators such as positivity rates don't necessarily account for the overall spread, since it depends on testing capacity. Moreover, we have chosen incidence rather than COVID-19 confirmed deaths because the latter is a lagged variable that could make it difficult to examine the relationship of COVID-19 spread with policies and economic outcomes. (In any case, we included indicators of COVID-19 deaths in S1 File that indicates that such indicators would lead us to similar conclusions.) We have used daily COVID-19 data reported by the New York Times.

This indicator is calculated as follows. First, raw data on COVID-19 incidence for each day and state is gathered and the seven-day moving average is calculated. The moving average is then divided by the 2019 census estimate of the state population [17], resulting in the seven-day moving average of new confirmed infections of COVID-19 per 100,000 people used in our analysis.

This indicator includes both confirmed and probable cases as defined by the Council of State and Territorial Epidemiologists (CSTE) [18] and the New York Times. The definition of confirmed cases includes those tested positive in a lab and reported by a federal, state, or territorial government. The New York Times definition of probable cases includes individuals who didn't have confirmed tests but were evaluated by health officials using guidelines created by state and federal government health departments. Antibody tests are not sufficient evidence on their own to constitute a counted case, and asymptomatic cases would not be included in counts unless a lab test was conducted [19].

Moreover, we have calculated three derived indicators. The first is the peak number of new COVID-19 infections in every state, defined as the maximum seven-day average of COVID-19 incidence between the start and end date of the lockdown phase. (Section 3.3 below describe how data on lockdown start and end in each state was gathered and standardized.)

The second indicator is the rate of change in new case infections, defined as the seven-day moving average of the rate of change in new cases in each state. The rate of change in day $t$ for state $i$ is defined as the derivative of daily COVID-19 incidence $inc(t)$:

$$r_i(t) = {}^{d\,inc(t)}\!/_{dt}$$

The third derived indicator is the duration, in days, of the lockdown phase in each state.

## 3.2. Economic data, including consumer spending

We have included daily data on consumer card spending, time spent at residential, work and other locations as an estimate of mobility, small business activity, as well as weekly data on unemployment insurance claims and job postings made available by Harvard University [13, 20]. We have also collected daily travel data from the Bureau of Transportation Statistics. Moreover, we have used data collected weekly or biweekly by the U.S. Census Household Pulse Survey [17] on reported food and housing insecurity, mental health and internet availability, and data collected monthly by the U.S. Census Current Population Survey [21] on weekly hours worked by gender and race, as a proxy estimate of gender and race inequality with respect to the pandemic impact.

Among the economic and mobility variables examined, we have chosen an indicator of consumer card spending for the analysis of economic activity in this paper, which is an indicator from Opportunity Insights of the de-seasonalized, seven-day average change (relative to January 2020) in consumer card spending at each state.

This indicator is calculated for each day $t$ as follows, based on Chetty et al. [13]. First, raw data on consumer card spending for each day $s(t)$ is taken as the 7-day moving average of card spending. The moving average is then seasonally adjusted by dividing $s(t)$ by the value of the corresponding day in 2019 $t^{2019}$:

$$s_{des}(t) = {s(t)}/{s(t^{2019})}$$

Finally, the seasonally-adjusted value $s_{des}(t)$ is used to calculate the change in consumer card spending relative to January 2020 as

$$spend(t) = {s_{des}(t)}/{\overline{s_{des}(t)}}$$

where $\overline{s_{des}(t)}$ is the average spending over the January 4–31 reference period. The variable *spend* (*t*) is the indicator used in this paper. This indicator was obtained by Opportunity Insights from credit and debit card transactions processed by Affinity Solutions, which represents about 10 percent of total consumer card spending in the U.S. The sources and calculations of this and other indicators from Opportunity Insights are described in detail in [13], which found that $spend(t)$ closely follows the trends in consumer card spending during the pandemic. We have chosen an indicator of consumer spending for the analysis of economic activity in this paper because for most of the states examined the other variables of economic activity (unemployment, business activity, etc.) represent similar trends as consumer card spending. Moreover, consumer card spending has been found to be a reasonable proxy for consumer spending during the pandemic and its impact on the U.S. economy (not including housing expenses such as rent or mortgages, or durable goods such as automobile purchases) [13].

### 3.3. Classification of state policy interventions

After gathering consumer card spending and COVID-19 data, we then classified policies into categories denoting the level of restrictions over time in each state at each point over time. The policies used to make these categorizations included state mandates on business closings or re-openings, social distancing, group gathering size guidelines and other health related policies on a state by state basis. We then matched the timeseries of COVID-19 spread and economic variables with policy interventions enacted in each state. Since each state differs on the timing and exact nature of each intervention, we applied a manual standardization procedure in order to be able to compare policy interventions across states and their effects on COVID-19 infections and consumer card spending. The procedure consisted of identifying state and/or local policy interventions in state government websites and then manually mapping them into the phases recommended in the "Guidelines for Opening Up America Again" released by the federal government [3]. In other words, each policy intervention has been classified either as "shelter in place" ("lockdown") or one of the three reopening phases (1, 2 or 3) from the White House/CDC Guidelines. This step was performed for 37 states (see Fig 3). These are the states where we have found enough data to enable a clear identification of lockdown and phased reopening, which we believe are representative of the situation of all U.S. regions with respect to the public health and economic outcomes of the COVID-19 pandemic. Consumer card spending and COVID-19 data for all 50 states are shown in S1–S3 Figs in S1 File.

### 3.4. Correlation between COVID-19 incidence and consumer card spending

Pearson correlation coefficients between daily card spending $spend(t)$ and COVID-19 new confirmed infections per day were calculated for each state and the period between March 13

and August 9, 2020, in order to assess the degree of linear relationship between COVID-19 spread and economic recovery. This period is selected because lockdown orders were implemented after March 13 for all 37 states where data about state policies were clearly available, and none of those states were experiencing a second major peak of COVID-19 incidence before the first week of August. Therefore, the correlations and analysis in this paper focus on the first wave of infections in the U.S.

## 3.5. Causal inference

Besides the correlation between COVID-19 incidence and consumer card spending, we also discuss the effect of state reopening policies on this spending indicator. To this effect we employ the difference-in-differences (DiD) estimator [22, 23] to quantify the average effect in consumer card spending in states that reopened in a certain period, compared to the average effect in states that did not reopen in the same period. This is frequently called the average treatment effect (ATE) and DiD is a popular method employed to study the effect of policies enacted in certain groups (e.g. states) but not in others.

The magnitude and statistical significance of the ATE are estimated as follows. We used the same time of reopening and states that reopened at that time identified by Chetty et al. [13]– South Carolina, Alaska, Georgia, Minnesota, and Mississippi, which reopened in the week between the April 20[th] and 27[th] 2020. The effect of reopening in these states is then compared either to the other 45 states as controls, or the 21 states considered in Chetty et al. as similar to the states that reopened in the April 20[th]-27[th] period—California, Connecticut, Delaware, Florida, Hawaii, Illinois, Indiana, Louisiana, Maryland, Massachusetts, Missouri, Nebraska, New Jersey, New Mexico, New York, Oregon, Pennsylvania, South Dakota, Virginia, Washington, and Wisconsin. We have chosen these two groups of states for two reasons. The first is to analyze groups of states different from the examples discussed in Sections 4.1, 4.2, and 4.3, in order to demonstrate that the discussion is not specific to the states illustrated. Second, this grouping allows us to directly compare our results with those reported by [13].

Let $\overline{Y^{reopened,before}}$ be the average change in consumer card spending $spend(t)$ for the 5 states that reopened, before the reopening period (see 3.2 for the definition of $spend(t)$). Let $\overline{Y^{reopened,after}}$ be the average $spend(t)$ for the states that reopened, after the April 20[th]-27[th] period. Let $Y^{control,before}$ and $Y^{control,afte}$ be the corresponding pre-reopening and post-reopening average for the 21 states that did not reopen in the April 20[th]-27[th] period. The average change in $spend(t)$ for the states that reopened is $\overline{Y^{reopened,after}} - \overline{Y^{reopened,before}}$, and the average change in $spend(t)$ for those that did not reopen is $\overline{Y^{control,after}} - \overline{Y^{control,before}}$. The DiD estimator $\hat{\beta}_1^{DiD}$ is the average change in $spend(t)$ for the states that reopened mins the average change in $spend(t)$ for the states that did not reopen [22]:

$$\hat{\beta}_1^{DiD} = \left( \overline{Y^{reopened,after}} - \overline{Y^{reopened,before}} \right) - \left( \overline{Y^{control,after}} - \overline{Y^{control,before}} \right)$$

The DiD above is estimated as regression of a panel with 26 states (5 reopening and 21 control states) and two time periods (before and after). For each state, the "before" data point is the average of the daily $spend(t)$ over three weeks before the April 20[th]-27[th] reopening, and the "after" data point is the average of $spend(t)$ over three weeks after reopening. This specification is recommended by Bertrand et al. [23] as a correction to avoid understating the standard errors of $\hat{\beta}_1^{DiD}$. Such correction is necessary because of significant autocorrelation in $spend(t)$, which is shown in S7 Fig in S1 File.

Several values of $\hat{\beta}_1^{DiD}$ are estimated. To capture some of the effect of heterogeneity among states in the reopening and control groups, $\hat{\beta}_1^{DiD}$ is estimated with and without additional

regressors—political majority (Republican or Democrat) [24] and median household income per state reported by the U.S. Census [25] (Table H-8, 2018). Moreover, $\hat{\beta}_1^{DiD}$ is estimated considering two versions of *spend*(*t*). One is the overall change in consumer card spending (relative to January 2020). The second version of *spend*(*t*) is the change in card spending by consumers living at zip codes in the top quartile of median income [20].

## 4. Results

In this section we describe the variation of COVID-19 incidence and consumer card spending over time. We also present the correlations between COVID-19 incidence and consumer card spending, and the relationship between these variables and state restrictions. The main results are shown in Figs 1–4. Those figures highlight the results of four states that serve as examples: New York, Michigan, California and Arizona. These examples were chosen because they differed both in the duration of lockdowns and timing of reopening, and in the evolution of the spread of SARS-CoV-2. These states are illustrative of trends in most of the states we analyzed and we discuss how these states represent the general trends amongst the U.S. overall. Trends and correlations for all U.S. states are shown in S1–S6 Figs in S1 File.

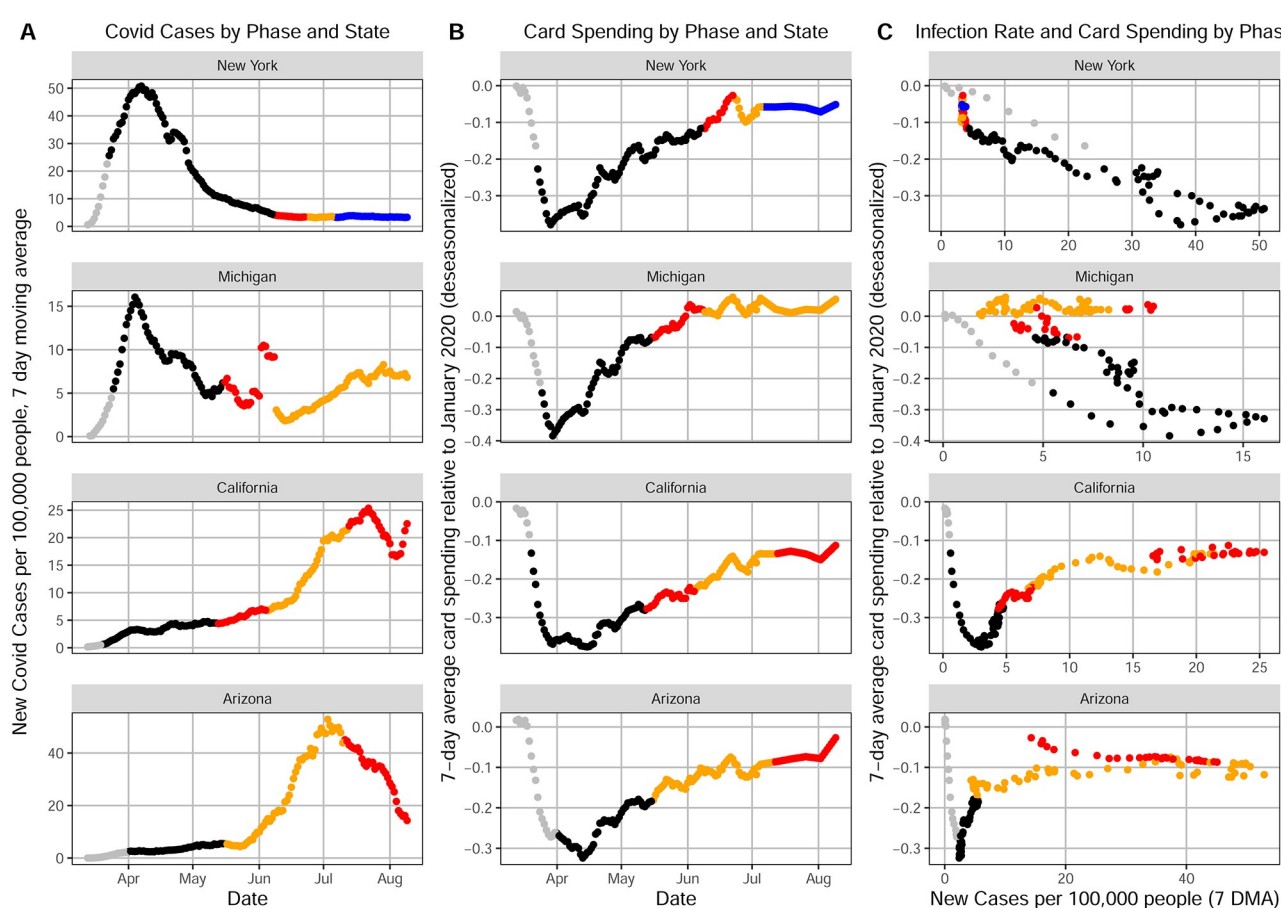

**Fig 1.** Temporal patterns between March 13 and August 9, 2020 for New York, Michigan, California and Arizona of (A) seven-day moving average of new confirmed daily COVID-19 cases per 100,000 people over time. (B) seven-day moving average of de-seasonalized change in consumer card spending (relative to January 2020) over time. (C) Consumer card spending as a function of COVID-19 incidence rate. The colors represent our classification of state policies into lockdown restrictions and the three phases of the White House/CDC reopening guidelines.

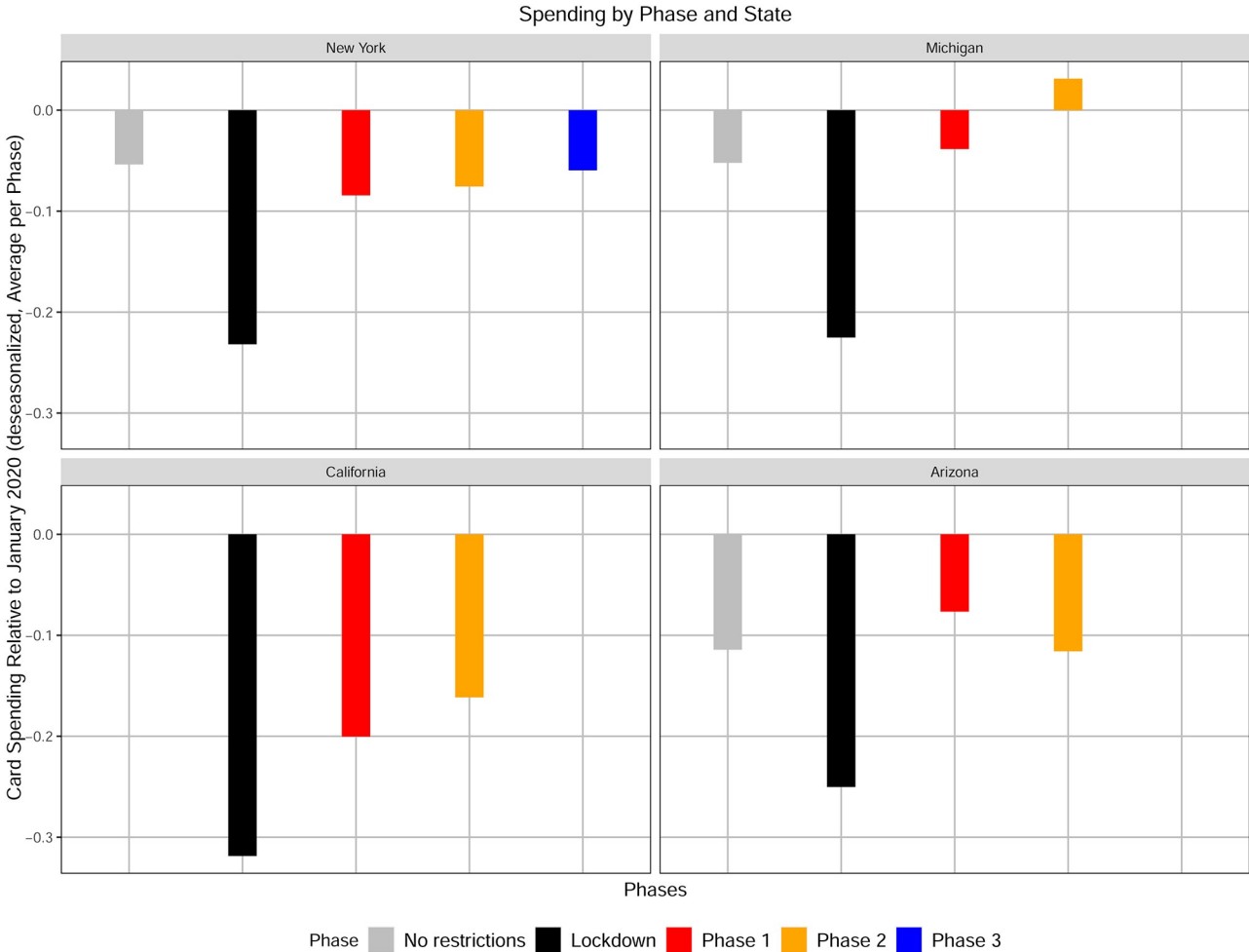

**Fig 2. Average change in consumer card spending per phase in New York, Michigan, California and Arizona.** Each bar represents the average change in consumer card spending (de-seasonalized and relative to January 2020) observed in the lockdown (black) and each phase of reopening defined by the White House/CDC guidelines. Each data point in the averaging represents the seven day moving average of the change in consumer card spending, de-seasonalized and relative to January 2020, in each state.

## 4.1. Temporal patterns of COVID caseload and policy response

Patterns of evolution in COVID-19 incidence were different in different states, as illustrated in Fig 1 for New York, Michigan, California and Arizona. Fig 1A shows the evolution of COVID-19 incidence over time, where each point corresponds to the daily count of new COVID-19 cases in the state (7-day moving average of new confirmed cases per 100,000 people). The colors of the points represent our manual classification of state restriction and reopening policies according to the White House/CDC guidelines [3]. The black points represent the lockdown mandates, while the red, orange and blue points represent phases 1, 2 and 3 of reopening, respectively, according with the interventions enacted in each state. S1 Fig in S1 File shows COVID-19 incidence over time for all 50 states.

COVID-19 incidence rates started and evolved differently with several states experiencing a relatively early peak of infections and other states having cases grow only at a later time. While New York and Michigan experienced a clear peak followed by a fast decline in the number of

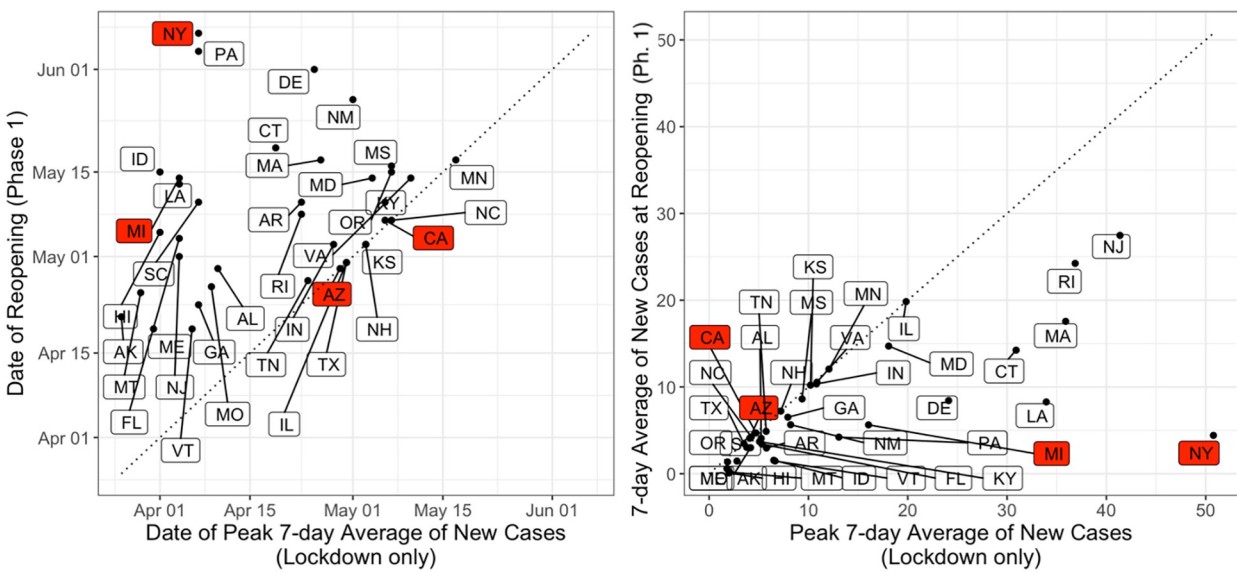

**Fig 3.** (A) Date of reopening to Phase 1 as a function of date of peak 7-day average of COVID-19 incidence during the lockdown, for the states where we classified phase information according to the White House/CDC guidelines (MI, NY, CA, AZ highlighted in red). (B) 7-day average of COVID-19 incidence per 100,000 people at reopening to Phase 1 as a function of peak 7-day average of COVID-19 incidence per 100,000 during the state lockdown, for the states examined.

new confirmed cases per day, California and Arizona exhibited a monotonic growth in the number of confirmed infections through most of the observation period.

Our data shows that lockdown policies in all states were quick to mimic those of states like New York where COVID-19 incidence rates were relatively high early in the first wave of the

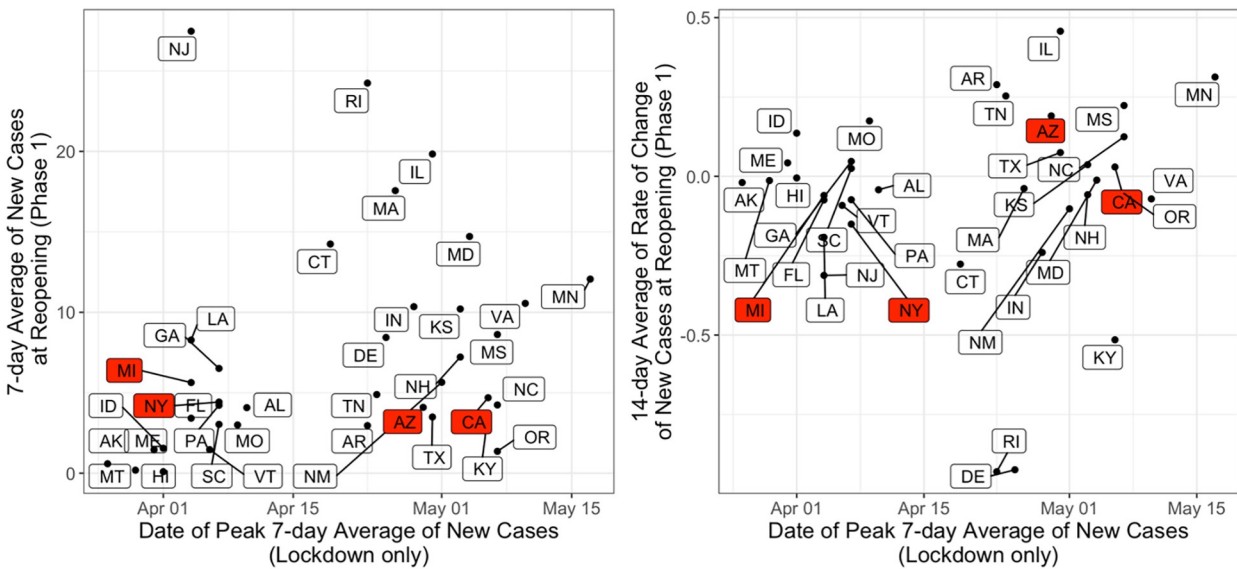

**Fig 4.** (A) 7-day average of COVID-19 incidence per 100,000 people at reopening to Phase 1 as a function of date of peak 7-day average of COVID-19 incidence per 100,000 people during the lockdown, for the states with phase information categorized. (B) 7-day average of rate of change of COVID-19 incidence per 100,000 people at reopening to Phase 1 as a function of date of peak 7-day average of COVID-19 incidence during the state lockdown.

pandemic. In fact, all 37 states where we obtained data on restriction and reopening policies implemented a lockdown at dates ranging from 3/19 to 4/7. On March 19th, California was the first state to implement the orders, and South Carolina was the last on April 7th.

However, it is evident that many states began to reopen at different paces into phases 1, 2, and 3 with respect to the level of COVID-19 in each state. Despite heterogeneity in states' incidence rates, many states moved forward with reopening decisions, as seen in Fig 1A. States like New York and Michigan, amongst others, reopened only when incidence rates were clearly decreasing. Not only was the incidence decreasing in Michigan and New York, but it was also far below the peak incidence rate during the lockdown for each of the respective states. Moreover, others found that lockdowns helped reduce the spread of COVID-19. For example, Amuedo-Dorantes et al. [8] noted that mortality is not uniform across states and in the U.S. Northeast (Connecticut, Maine, Massachusetts, New Hampshire, Rhode Island, Vermont, New Jersey, New York, and Pennsylvania), which experienced aggressive early spread of SARS-CoV-2, accelerating lockdown implementation by the equivalent one day could have decreased COVID-19 deaths by 7.6 percent.

On the other hand, states like Arizona and California advanced in reopening only to then experience marked increases in COVID-19 incidence rates. These states both reopened at or close to the highest incidence rate throughout the lockdown period. This type of policy action was not unique to Arizona and California but was similar to several other states such as Florida, North Carolina or South Dakota. Thus, it leads to the question of whether it gave consumers a false sense of health security and confidence to resuming normal activity.

## 4.2. Economic patterns of consumer behavior

We examine the consumer card spending activity as a function of time and policy response phase for the example states in Fig 1B, and for all 50 states in S2 Fig in S1 File. Each daily point represents the seven-day moving average of change in consumer card spending in the state, de-seasonalized and relative to January 2020 as a pre-pandemic reference. (A value of zero means that card spending in that day is equivalent to the level in January 2020. The details about the definition of this variable are described in the Method Section.) We find a universal trend of sharp spending reductions right after COVID-19 arrived in the U.S. followed by steady increases almost unanimously throughout the states. Consumer card spending started to go down even before lockdown orders were implemented and bottom in most of the states in late March-mid April. After states entered the lockdown, there was a clear upward trajectory of consumer card spending as time passed for the four states shown in Fig 1B. The sharpest recovery in spending takes place in the period between the lockdown and phase 1 of reopening. Additionally, recovery in card spending continues from phase 1 through phase 2 for most states, although such a trend is less clear for phase 3. This contradicts patterns of COVID-19 incidence in different states, which are further discussed below.

## 4.3. Correlation of policy response and consumer economic behavior

Fig 1A and 1B show a pattern of COVID-19 incidence that is different in each state, while the decline and recovery in consumer card spending is relatively homogeneous across states. Because of this we observe contrasting trends. Consumer card spending recovered when COVID-19 incidence rates were declining in some states, whereas in others spending and COVID-19 increased together. Such a contrast is shown in Fig 1C, which displays card spending as a function of new infections for New York, Michigan, California and Arizona. It shows a negative correlation between COVID-19 incidence and consumer card spending for some states but a positive correlation for others. In New York and Michigan, we observe that

consumer card spending is negatively correlated with COVID-19 daily incidence rates. The Pearson correlation is −0.95 for New York and −0.66 for Michigan. This suggests that consumer card spending declined when the pandemic was at its worst and then gradually recovered after the peak of the first wave. These observations are in a sharp contrast with patterns observed for California and Arizona, both of which display a positive correlation (0.72 and 0.59, respectively) between consumer card spending and COVID-19 incidence, revealing that consumers spent more when COVID-19 incidence is highest in these states. These patterns are not specific to the states of California and Arizona but are also observed in other states, as seen in S3 Fig in S1 File. Besides, consumer card spending on food and groceries are relatively less correlated with COVID-19 incidence in the example states (as shown in S5 Fig in S1 File), indicating that spending on non-food categories varied more with the spread of SARS-CoV-2 than food-related categories.

We examined whether the pattern of correlation between consumer card spending and COVID-19 (negative correlation for New York and Michigan and positive for California and Arizona) are specific to incidence rates, and we found that the same pattern applies when COVID-19 deaths are used instead of incidence (as shown in S6 Fig in S1 File).

We examine in more detail the relationship between policy interventions and consumer card spending, which is illustrated in Fig 2. For this graph we average the daily change in spending for each of the four phases (lockdown and phases 1, 2, and 3) in each of the example states. Consumer card spending decreases sharply and rebounds gradually in all states examined, regardless the differences in COVID-19 incidence rates observed across the states.

The pattern of relative similarity in consumer card spending across states suggests that government mandates are the primary factor in the consumer's spending decisions over time, motivating us to examine the discrepancy between government mandates and the progression of the COVID-19 pandemic.

## 4.4. Effectiveness of policy response

To better understand the effect of state interventions on COVID-19 spread and economic recovery and resilience, we examine the timing of reopening interventions and how they overlap with trends of COVID-19 incidence rates in each state. Regarding lockdown orders, there was relatively little temporal variation across states. In contrast, there is significant temporal variation in states transitioning to reopening from the lockdown. There is also wide variation in the trends regarding COVID-19 incidence when states decided to reopen. This heterogeneity in time and COVID-19 incidence rates is evident in Fig 3. Fig 3A shows the date each state transitioned from the lockdown to phase 1, as a function of the date with the highest seven-day moving average of COVID-19 incidence during the lockdown. In case the peak was not observed during the lockdown phase, we consider the lockdown peak date to be equal to that of the reopening date. Complementary to reopening dates, we analyze the daily infection rates observed at the reopening and compare them to those at the COVID-19 peak of incidence, which is shown in Fig 3B. Being further below the diagonal line of both Fig 3A and 3B represents better timing for a state's reopening in terms of mitigating the spread of COVID-19. A state reopening before the incidence of COVID-19 reaches the peak is observed places the state on the diagonal, which is the case of California and Arizona. Conversely, one expects a state reopening to take place long after the COVID-19 peak when the daily incidence rate is substantially lower that that at the peak, placing the state far from the diagonal in Fig 3A and 3B, such as New York, Michigan, and Massachusetts.

Among the states shown, Vermont and Maine are the first two to reopen on April 20[th] and New York is the last state on June 6[th]. The mean of new daily cases per 100,000 people was 8.07

on the dates that states transitioned to phase 1, with significant dispersion. The standard deviation of 7.83, with New Jersey having 27 new daily cases per 100,000 people at reopening. This reflects the differences in reopening strategy from state to state, with respect to whether they decided to reopen when COVID-19 incidence rates were increasing or decreasing. This is further illustrated in Fig 4. Fig 4A shows the daily incidence at reopening, and Fig 4B shows the rate of change in daily incidence. In both Fig 4A and 4B, r the date when incidence peaked (in the lockdown period) is shown in the horizontal axis. The horizontal axis illustrates the variation in the dates when states experienced the first peak of incidence rates. Perhaps more importantly, while Fig 4A shows that the majority of states decided to reopen when COVID-19 incidence rates where no higher than 10 new cases per day, Fig 4B shows wide variation with respect of the trend of incidence at the time of reopening. States such as New York, Michigan, and several others decided to start reopening when the rate of change was negative, meaning that daily incidence was decreasing. On the other hand, many other states such as Arizona and California started reopening when the rate of change was positive, meaning that daily COVID-19 incidence was increasing during reopening.

**4.4.1. Causal inference with difference-in-differences estimator.** Our estimates of the average treatment effect of state reopening are as follows. Fig 5 summarizes the ATE for overall consumer card spending, considering SC, GA, MN, MS, and AK as the states that reopened in the week between April 20th and April 27th 2020 [13], and all 45 remaining states as controls. The green line shows the change in the average $spend(t)$ for the control states from the 2 weeks before to after reopening is 8.02% (the 2-week period before and after is as in Chetty et al.'s [13], but this is varied later). On the other hand, for the reopening states the average $spend(t)$

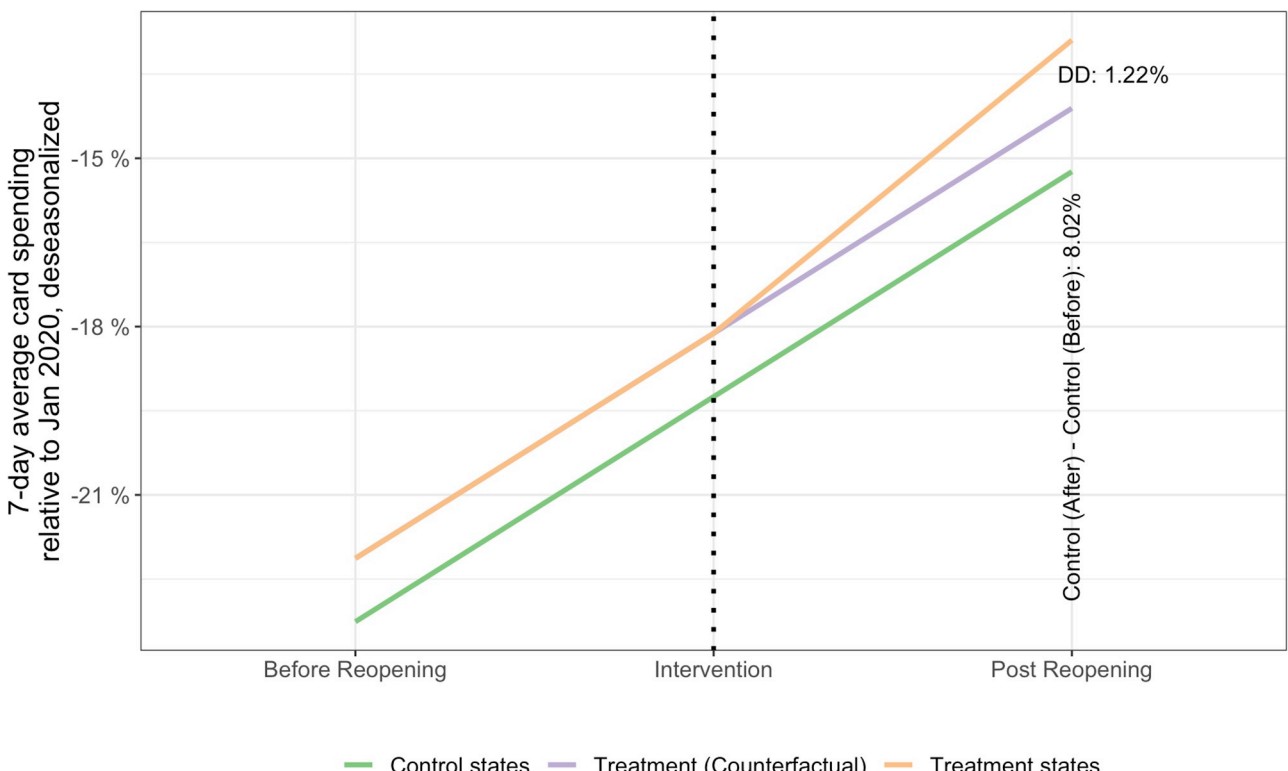

**Fig 5. Average $spend(t)$ (overall change in consumer card spending relative to January 2020) of states that reopened in the week of April 20th-27th 2020, compared to states that did not reopen.** The groups of states "treatment" and "control", as well as the periods "Before Reopening" and "Post Reopening" are described in section 3.5. $\overline{Y^{control,before}} = -23.3\%$, $\overline{Y^{control,after}} = -15.24\%$, $\overline{Y^{reopened,before}} = -22.1\%$, $\overline{Y^{reopened,after}} = -12.9\%$.

changed 1.22% more than the amount these states would have if they had the same recovery as the control group (orange line). This change is the DiD estimate of the average treatment effect. (The parallel trends assumption of difference-in-differences regressions states that the control group and the treatment group follow the same trends in the outcome variable absent the treatment, illustrated by the purple line.) We assume parallel trends in consumer spending between the two groups before the reopening based on the homogeneity among the spending curves shown in S2 Fig in S1 File and the choice of groups by Chetty et al. Indeed, our estimate for the ATE is close to Chetty et al.'s [13] for the same period and reopening states (even considering a different groups of control states—all remaining 45 states in Fig 5). Nevertheless, we note the relative magnitude of this effect. The ATE corresponds to an elasticity of 1.22% / 8.02% = 15.2%. In other words, we estimate that the recovery in consumer card spending in the states that reopened in late April is 15.2% more than the recovery in the other 45 states.

Table 1 shows the results of the panel regression for the model described in section 3.5. The first row shows the ATE estimate $\hat{\beta}_1^{DiD}$ (i.e. the effect of state reopening). Column (1) shows the results controlled for time (week) and state fixed effects. The estimate of reopening effect is statistically significant at the 5% level. Column (2) adds the political majority as an additional regressor. The coefficient of this regressor is not statistically significant. Moreover, it increases the standard error of the reopening effect, which is a possible indication of multicollinearity (political majority correlated with reopening decisions). Similar results are obtained with county median household income (column 3) and county population density (column 4) as additional regressors to control for income and urbanization conditions that may influence behavior as discussed in [26]. (Moreover, S8 and S9 Figs in S1 File indicate that residuals are normally distributed and approximately homoscedastic, and we calculated the correlation between each regressor and the residuals to be less than 0.01. These suggest that the estimates of the regression coefficients are unbiased and consistent.)

Columns (5) and (6) show results similar with the model in (1) but considering different numbers of weeks before and after the reopening week. The similarity between the results suggests that the different between the states that reopened in late April and the other states persisted beyond the first two weeks. (In any case, we didn't considered data prior to April 2020, because that would include the widespread decline in consumer card spending following the lockdown orders in March, shown in Fig 1 and S2 Fig).

Column (7) shows results similar with the model in (1) but considering only the 21 control states used in [13]. This model shows a larger effect for the effect of reopening on consumer card spending than in (1), but the standard error is also larger. Columns (8) and (9) are similar to the models in (1) and (7), respectively, but using state-level observations. The model with state-level observations for all states (column 8) results in a non-statistically significant effect of reopening.

We also estimate the ATE of state reopening on consumer card spending by top income families in the state. Fig 6 summarizes the ATE for consumer card spending, but this time for cards with addresses at zip codes with median household income in the top quartile for each state. The change in the average *spend*(*t*) for the control states from the 2 weeks before to after reopening is 8.02% (green line). On the other hand, for the reopening states the average *spend*(*t*) changed 1.46% more than the amount these states would have if they had the same recovery as the control group (orange line). The latter is the DiD estimate of the average treatment effect on card spending for higher income families. The relative magnitude of this effect is higher than for overall spending. The ATE corresponds to an elasticity of 1.46% / 8.02% = 18.2%. In other words, we estimate that the recovery in high income consumer card spending in the states that reopened in late April is 18.2% more than the recovery in states that did not reopen

**Table 1. Difference-in-differences model results for card spending.**

| | Dependent variable: 7-day average card spending relative to Jan 2020, deseasonalized | | | | | | | | |
| | (1) | (2) | (3) | (4) | (5) | (6) | (7) | (8) | (9) |
|---|---|---|---|---|---|---|---|---|---|
| Effect of reopening policy | 1.215** (0.515) | 1.148** (0.565) | 1.214** (0.537) | 1.205** (0.516) | 1.129* (0.671) | 1.200* (0.629) | 2.043** (0.825) | 0.539 (0.409) | 1.389** (0.650) |
| Republican majority | | 0.449 (0.385) | | | | | | | |
| Median income | | | -0.0003 (0.013) | | | | | | |
| Population density | | | | -0.0001 (0.0003) | | | | | |
| Time indicator | 8.021*** (0.393) | 7.684*** (0.334) | 8.039*** (1.069) | 8.042*** (0.401) | 10.004*** (0.480) | 11.838*** (0.422) | 9.090*** (0.679) | 8.355*** (0.294) | 9.443*** (0.477) |
| CI 95% | (0.205, 2.225) | (0.042, 2.255) | (0.163, 2.266) | (0.194, 2.217) | (-0.186, 2.444) | (-0.032, 2.432) | (0.427, 3.659) | (-0.263, 1.340) | (0.116, 2.662) |
| N | 8,695 | 8,695 | 8,695 | 8,695 | 12,173 | 15,651 | 6,762 | 250 | 182 |
| F Statistic | 1,251.168*** (df = 2; 6954) | 834.584*** (df = 3; 6953) | 833.992*** (df = 3; 6953) | 834.173*** (df = 3; 6953) | 2,473.784*** (df = 2; 10432) | 4,011.202*** (df = 2; 13910) | 1,257.678*** (df = 2; 5794) | 135.561*** (df = 2; 198) | 154.165*** (df = 2; 154) |

Notes:

***Significant at the 1 percent level.

**Significant at the 5 percent level.

*Significant at the 10 percent level.

The table shows the results for panel regressions controlled for time (week) and state fixed effects. The first row of the table shows the DiD estimate (with standard error in parenthesis) of the effect of state reopening between April 20th and 27th 2020 on the percent change in consumer card spending (relative to January 2020). SC, GA, MN, MS, AK are considered as the treatment states [13]. All other 45 states are considered as control states. The "time indicator" is a variable that equals 1 after reopening. CI 95% is the 95% confidence interval for the effect of reopening on consumer card spending. N is the number of observations. Standard errors are clustered at the state level.

Column (1) shows the DiD estimates considering 2 weeks before and 2 weeks after the reopening week. Observations are at the county level. Column (2) shows the addition of the county-level political majority (Republican = 1) using data from [27]. Since political majority is fixed during the period of interest, the effect shown is the interaction between the majority and time indicators. Likewise, column (3) shows results for the model in (1) with the addition of county median household income (in USD thousands) using data from [28], and column (4) shows results for the model in (1) with the addition of population density per km² using data from the US Census ACS. Column (5) shows results similar with the model in (1) but considering 3 weeks before and after the reopening week. Likewise, column (6) considers 3 weeks before and 5 weeks after the reopening week. Column (7) shows results similar with the model in (1) but considering only the 21 control states used in [13]. Columns (8) and (9) are similar to the models in (1) and (7), respectively, but using state-level observations.

in the same period. However, Table 2 shows the results of the panel regression considering card spending by high income families, and the ATE estimate $\hat{\beta}_1^{DiD}$ is *not* statistically significant, even with the addition of additional regressors (columns 2–4), or different periods before and after the late April reopening (columns 5–6). This is possibly because the dataset has state-level data, but not county-level data (as in Table 1), for consumer card spending of households from top income zipcodes. (Note that the standard errors in Table 2 are considerably higher than the standard errors in Table 1.) The only exception is column (7), which coefficient of the effect of reopening is significant at the 10% level. This results considers only the 21 states as controls used in [13].

## 5. Discussion and conclusions

We have analyzed publicly available data to assess how consumer card spending, as an important indicator of economic activity, varied based upon state decisions to enter and exit

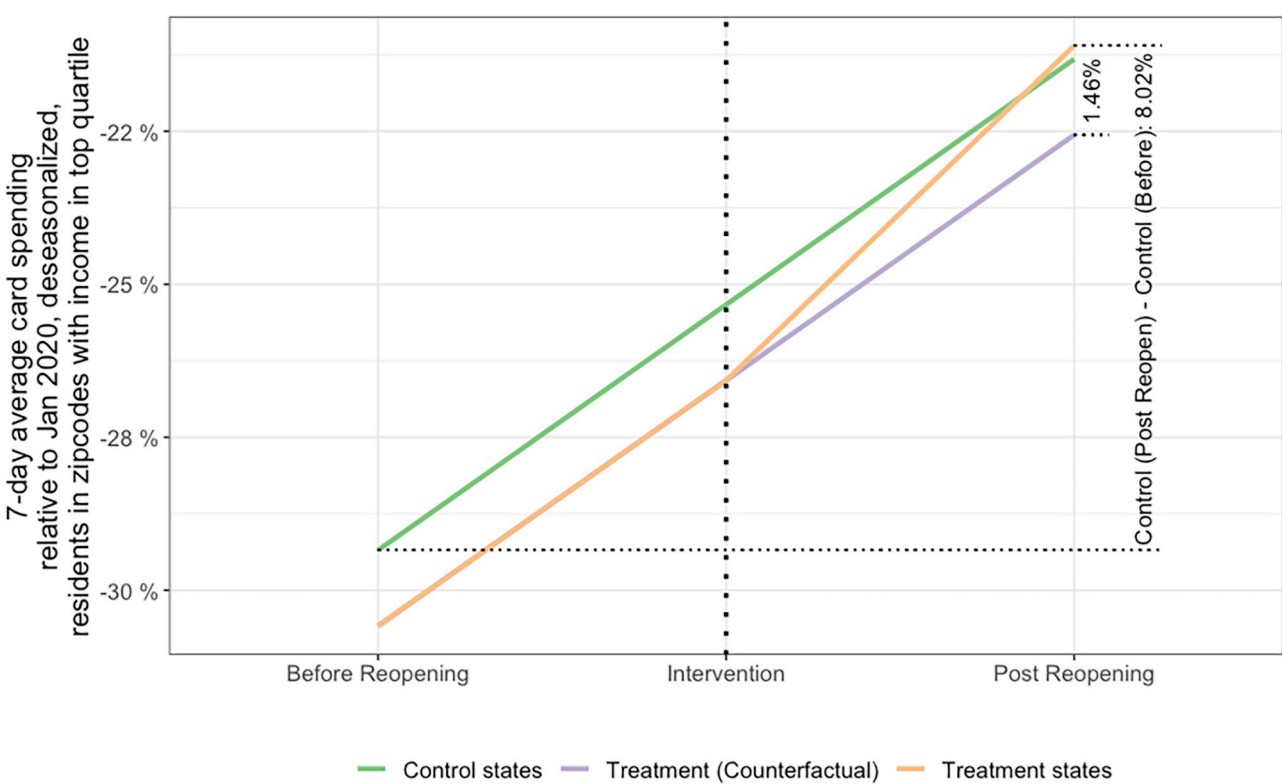

**Fig 6. Average *spend(t)* (consumer card spending of cards with address at zip codes with median household income in the top quartile for each state) of states that reopened in the week of April 20th-27th 2020, compared to states that did not reopen.** The groups of states "treatment" and "control", as well as the periods "Before Reopening" and "Post Reopening" are described in section 3.5. $\overline{Y^{control,before}} = -29.3\%$, $\overline{Y^{control,after}} = -21.32\%$, $\overline{Y^{reopened,before}} = -30.58\%$, $\overline{Y^{reopened,after}} = -21.11\%$.

lockdowns and trends in COVID-19 incidence. First, we find little consistency across states regarding the trend in COVID-19 incidence when lockdown and reopening decisions were made. While some experienced high spread in the beginning of the pandemic, cases grew relatively later in other states. Despite this difference, lockdowns were mandated in most states within a relatively short time window, probably following the actions of early-hit states such as New York rather than being triggered by situation of COVID-19 at the time in the state. For the states which we have reopening information, the majority of them decided to reopen when COVID-19 incidence was below 10 new cases per day, which at the time was considered a situation of "minimal community spread" [29]. However, states differed with respect to the *trend* of COVID-19 incidence preceding reopening. New York, Michigan, Massachusetts and other states decided to cautiously reopen when new daily cases were decreasing, while California, Arizona and many others started reopening with growing COVID-19 incidence. In those states, policymakers may have relied only on the low incidence levels and may not have had full visibility of the upward trends in incidence or in projecting the implications of different speeds of reopening on the health security of the people in their states.

We have compared trends of COVID-19 spread and each state's lockdown and reopening mandates with data on consumer card spending in order to assess how differences in policy timing impacted economic recovery. While the pandemic evolution and interventions varied, consumer card spending increased almost uniformly across the states examined. Correspondingly, during the reopening process people spent progressively more money as COVID-19

**Table 2. Difference-in-differences model results for card spending in high-income zipcodes.**

| | Dependent variable: 7-day average card spending relative to Jan 2020, deseasonalized, for households in zipcodes with income in top quartile in state | | | | | | |
|---|---|---|---|---|---|---|---|
| | **(1)** | **(2)** | **(3)** | **(4)** | **(5)** | **(6)** | **(7)** |
| Effect of reopening policy | 1.456 (1.452) | 1.133 (1.546) | 1.164 (1.361) | 1.407 (1.503) | 0.434 (0.898) | 0.103 (0.920) | 1.445* (0.819) |
| Republican majority | | 1.454 (1.029) | | | | | |
| Median income | | | -0.056 (0.051) | | | | |
| Population density | | | | -0.001 (0.005) | | | |
| Time indicator | 8.025*** (0.596) | 7.184*** (0.576) | 11.657*** (3.652) | 8.114*** (0.858) | 9.825*** (0.620) | 12.375*** (0.675) | 8.813*** (0.490) |
| CI 95% | (-1.390, 4.301) | (-1.898, 4.163) | (-1.503, 3.831) | (-1.540, 4.353) | (-1.326, 2.194) | (-1.701, 1.906) | (-0.161, 3.051) |
| N | 250 | 250 | 250 | 250 | 350 | 450 | 182 |
| F Statistic | 98.030*** (df = 2; 198) | 65.994*** (df = 3; 197) | 65.613*** (df = 3; 197) | 65.047*** (df = 3; 197) | 216.007*** (df = 2; 298) | 357.036*** (df = 2; 398) | 140.769*** (df = 2; 154) |

Notes:

***Significant at the 1 percent level.

**Significant at the 5 percent level.

*Significant at the 10 percent level.

The table shows the results for panel regressions controlled for time (week) and state fixed effects. The first row of the table shows the DiD estimate (with standard error in parenthesis) of the effect of state reopening between April 20th and 27th 2020 on the percent change in consumer card spending (relative to January 2020), deseasonalized, for households in zipcodes with income in the top quartile in state. SC, GA, MN, MS, AK are considered as the treatment states [13]. All other 45 states are considered as control states. The "time indicator" is a variable that equals 1 after reopening. CI 95% is the 95% confidence interval for the effect of reopening on consumer card spending. N is the number of observations. Standard errors are clustered at the state level. Observations are at the state level.

Column (1) shows the DiD estimates considering 2 weeks before and 2 weeks after the reopening week. Column (2) shows the addition of the county-level political majority (Republican = 1) using data from [27]. Since political majority is fixed during the period of interest, the effect shown is the interaction between the majority and time indicators. Likewise, column (3) shows results for the model in (1) with the addition of county median household income (in USD thousands) using data from [28], and column (4) shows results for the model in (1) with the addition of population density per $km^2$ using data from the US Census ACS. Column (5) shows results similar with the model in (1) but considering 3 weeks before and after the reopening week. Likewise, column (6) considers 3 weeks before and 5 weeks after the reopening week. Column (7) shows results similar with the model in (1) but considering only the 21 control states used in [13].

incidence decreased in some of the states we examined such as New York and Michigan, while new cases underwent sharp increases upon or after reopening in others (such as California and Arizona). Although this recovery in card spending might benefit many people and businesses who were negatively impacted by the pandemic when it comes to the jobs lost or decline in revenues, it reflects the heterogeneity in correlations between the level of COVID-19 in a state and the timing of its NPI.

Our observations are consistent with previous studies [13], and they suggest that the following hypothesis could be true: consumer card spending was not driven by the level of COVID-19 in a given state but by government mandates imposed by the state. We find that consumer card spending was strongly correlated with the reopening of a state, as mandated by state and local governments. Contrasting with our expectations, however, the finding that not all states reopened after COVID-19 incidence rates were under control suggest that the economic recovery might have been more related with reopening mandates than with the COVID-19 situation when those mandates were implemented. This is consistent with our quantitative estimate of the causal effect. We find that in states that reopened between the 20th and 27th of April, recovery in overall consumer card spending was 15% more than the recovery in states that did not reopen in the same period. The effect of reopening on consumer card spending among high income households is somewhat higher. For the same period, we found the

recovery in the states that reopened was 18% more than in the states that did not reopen. (However, we obtained mixed results regarding the statistical significance for high income households.)

What drove the nearly uniform rise in card spending throughout the states, if not a decline of COVID-19 incidence rates? Several factors may explain the uniformity in consumer card spending trends. One possibility is the implementation of lockdown orders in states with high COVID-19 incidence creating a transmission of fear to other state governments that proceeded to take similar policy action. Consequently, this domino-effect in policy implementation may have created spillover in the form of over-precautionary consumer behavior and fear in many states. This phenomenon could be a factor in the nationwide, rapid decline in card spending early on in the pandemic, even in states that experienced relatively lower peaks of COVID-19 incidence during the lockdown period [30]. This is consistent with the logic of Goolsbee and Syverson [15], who find that fear was an important factor in deterring people from spending. In contrast, state reopening decisions apparently led to card spending resurgence, despite levels of COVID-19 incidence rates. Mid-April was around the time that consumer card spending shifted from a sharp, negative downward trend to a gradual upward trend in the case of the four example states shown in Fig 1 and others. One factor may be the influence of media on consumer confidence. Fellows et al. [31] assert that decreases in mobility may have come from perceived risk and other factors such as media coverage, and Marzouki et al. [32] found that social media played an important role in the public perception of uncertainty through the evolution of the pandemic. Moreover, a decline in compliance to orders may have affected states differently. The University of Oxford have developed a "stringency index" by which states can be compared at their level of policy stringency [33]. According to this metric, New York began with as high of a stringency level when the pandemic began, and it slightly reduced its level of stringency beginning in June through the rest of the summer, reflecting their long lockdown period and the prolonged measures they took to make sure they did not reopen too early. Michigan began the pandemic at an average level of stringency and slowly reduced its policy stringency only after the first wave receded. The level remained relatively constant during the first wave. On the other hand, California's level of stringency remained constant until late May when there was a sudden drop, and Arizona began the pandemic at a slightly below average level of stringency, but quickly experienced a sharp decline almost immediately before the first wave. This could be explained by their immediate transition from lockdown to phase two of the reopening process.

Previous work may shed light on demographic differences determining stringency of restrictions. Amuedo-Dorantes et al. [8] take into account the local political alignment, i.e. which counties were republican or democrat. They found that NPI adoption speed was a less relevant factor in republican counties. In addition, Alexander and Karger found that while republican counties had slower adoptions of NPI, they had nearly identical responses in mobility once restrictions were adopted, indicating that county-level traits leading to policy differences were not symbolic of how populations would behave once policy was enacted [34]. Alternatively, Fan et al. [35] found that democrats were more likely to limit socializing and gathering, and they were more likely to take precautions like wiping groceries, washing their hands, and wearing masks. Furthermore, they found that controlling for political affiliation and news consumption eliminated statistical differences in COVID-19 related beliefs.

If COVID-19 incidence rates were high but state governments mandates were eased, it is likely that many people would have felt safer increasing mobility and resuming more normal activity, therefore, contributing to higher levels of consumer card spending. This would explain why policy mandates seemed to have such a domineering effect on consumer card spending, even over factors like COVID-19 incidence.

Using illustrative examples of states' strategies for reopening, we can get a glance into the heterogeneity across states in reopening decisions and outcomes. In an article published on April 14[th], the governor of California indicated six factors for modifying lockdown policies, four of which were related to health risks, one based on social distancing capacity, and the last was the ability of the state to reinstitute previously mandated health safety measures [36]. The governor placed a strong emphasis on the science and data when determining whether or not to reopen. California's lockdown orders succeeded in mitigating COVID-19 incidence, which by mid-April was the lowest among the 25[th] top percentile of most densely populated states [37]. However, the fact that they relied on science and data to inform their decisions did not by any means ensure optimal policy action on reopening. In fact, California opened when the incidence trajectory was upwards, and the end result was even higher incidence rates per capita. As a consequence, the governor, praised for his actions early in the pandemic, was later met with protests [38] and threatened with a recall. Possible reasons for reopening with increasing COVID-19 incidence rates include (i) decisions considered inconsistent or discordant with science and data; and (ii) vanishing popular support, perhaps caused by disagreement with political, non-profit and private organizations (e.g. disobedience of companies such as Tesla that openly refused to follow restrictions) [39, 40]. Such a decline in support for restriction measures was also observed in other states that replaced lockdown orders for testing and contact tracing as their primary strategies to contain the spread of SARS-CoV-2 [41]. Another example of a state with sharp increase in the spread of SARS-CoV-2 virus following the reopening is Arizona. Possible causes are believed to be related to premature reopening decisions combined with the lack of a statewide mask mandate [42].

Comparatively, New York represents a case where the population adhered strictly to lockdown policies for an extensive period of time. For example, nearly 90% of people in New York City believed that their city's restrictions were either "the right balance or not restrictive enough" [43]. Moreover, most adults from New York City admitted in the survey that they would not feel safe if lockdown orders and business closures were removed. Relative to Arizona, there was less pressure on the government to reopen businesses and lift lockdown orders quickly when New York was being hit by the first wave. Keeping measures in place may have been easier from a policy making standpoint due to public support.

Did these different courses of action between states result in different outcomes? Despite rising cases, California proceeded to reopen further to phase 2, which induced a higher rate of infection. This forced California to revert back to phase 1, eventually leading to the abating of the first wave of COVID-19 in California. Contrastingly, we see that New York, which did not veer far from its initial plan, fared much better when reopening their economy. Transitioning into phase 1 close to their target for the incidence rates, they did not experience an increase in COVID-19 incidence or prevalence. In fact, they avoided the summer spike in COVID-19 incidence that California, Arizona and several other states experienced, without seeing a significant spike in cases until the resurgence of COVID-19 on a national scale after October.

The findings and hypotheses discussed above inform scholars and policymakers on the response to future public health crises. In particular, understanding the timing of state decisions regarding reopening and the relationship between those mandates and the economic recovery may help policymakers estimate how both COVID-19 and government interventions improve affect economic recovery while minimizing the impact on public health. The idea of fear as a primary driver of the sharp reduction in consumer card spending during the emergence of the pandemic is supported by the data and related literature.

On the other hand, our main finding is that the *recovery* in consumer card spending following the first wave of the pandemic was more related to state decisions to reopen from the lockdowns than the trends in the spread of SARS-CoV-2 at the time. In this case, reopening

decisions are a powerful mechanism to influence population behavior, and the timing of these decisions may affect both the economic and the health outcomes in the current and future pandemics. This correlation explicitly indicates the importance of planning and implementing resilience in governance strategies to assure appropriate management of future pandemics [44].

## Supporting information

**S1 File.**
(PDF)

## Acknowledgments

The work is resulted from author's work at FEMA/HHS Region 1 COVID Task Force. The authors are grateful to Captain W. Russell Webster, USCG (Ret.), CEM, Region 1 Administrator, who requested these analyses in support of his decision making and to Gary Kleinman, HHS/ASPR Regional Administrator whose leadership inspired us. Our special thanks to Drs. Melissa Surette and Susan Cibulsky for leading the Data Analytics team. The views and opinions expressed in this article are those of the individual authors and not those of the U.S. Army or other sponsor organizations.

## Author Contributions

**Conceptualization:** Alexandre K. Ligo, Jeffrey Cegan, Benjamin D. Trump, Maksim Kitsak, Jesse Keenan, Igor Linkov.

**Data curation:** Alexandre K. Ligo, Emerson Mahoney, Andrew S. Jin.

**Formal analysis:** Alexandre K. Ligo, Emerson Mahoney, Benjamin D. Trump, Andrew S. Jin, Maksim Kitsak, Igor Linkov.

**Methodology:** Alexandre K. Ligo.

**Software:** Emerson Mahoney.

**Supervision:** Igor Linkov.

**Validation:** Jeffrey Cegan, Maksim Kitsak, Jesse Keenan, Igor Linkov.

**Visualization:** Alexandre K. Ligo, Emerson Mahoney, Andrew S. Jin.

**Writing – original draft:** Alexandre K. Ligo, Emerson Mahoney, Jeffrey Cegan, Benjamin D. Trump, Andrew S. Jin, Maksim Kitsak, Igor Linkov.

**Writing – review & editing:** Alexandre K. Ligo, Emerson Mahoney, Jeffrey Cegan, Benjamin D. Trump, Andrew S. Jin, Maksim Kitsak, Jesse Keenan, Igor Linkov.

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
