## [Decision Letter · Decision Letter 0]

21 Jul 2021

PONE-D-21-21306

Relationship among state reopening policies, health outcomes and economic recovery through first wave of the COVID-19 pandemic in the U.S.

PLOS ONE

Dear Dr. Ligo,

Thank you for submitting your manuscript to PLOS ONE. After careful consideration, we feel that it has merit but does not fully meet PLOS ONE’s publication criteria as it currently stands. Therefore, we invite you to submit a revised version of the manuscript that addresses the points raised during the review process.

We look forward to receiving your revised manuscript.

Kind regards,

Martial L Ndeffo Mbah, Ph.D

Academic Editor

PLOS ONE

Additional Editor Comments:

Regarding Reviewer #1 following comment:

Second, one of the big arguments is that it's not so much infections per se driving down consumption, but fear about transmission that depresses expectations and consumer confidence. What is your answer to that? One way is to compare states that are similar in political affiliation, but different in state policies. But, that goes back to #1 in needing controls and placebos.

I think that it is an important point to explore, but it will be more interesting to look at it from a different perspective than political affiliation.

Journal Requirements:

Reviewers' comments:

Reviewer's Responses to Questions

**Comments to the Author**

1. Is the manuscript technically sound, and do the data support the conclusions?

Reviewer #1: Partly

Reviewer #2: Yes

2. Has the statistical analysis been performed appropriately and rigorously? 

Reviewer #1: No

Reviewer #2: Yes

3. Have the authors made all data underlying the findings in their manuscript fully available?

Reviewer #1: Yes

Reviewer #2: Yes

4. Is the manuscript presented in an intelligible fashion and written in standard English?

Reviewer #1: Yes

Reviewer #2: Yes

5. Review Comments to the Author

Reviewer #1: Dear Authors:

Thank you for your work in this paper. I am in full agreement that state policies had much more to do with the fluctuations in consumption than the actual infection rate, but the execution of the statistical work in the paper needs an overhaul.

First, there is zero discussion of causal inference. If your goal is simply to say that consumption is better explained by state policies than by infections, then yes you can win that -- but that's already a fairly obvious point that we can see just by observing stories in the press. The trillion dollar question is how much the different policies have contributed to slower versus faster consumption growth, particularly in the length of time that the restrictions were imposed and their severity. But to do that, you at least need to control for confounding factors. Obviously there is only so much you can do -- there is too much time-varying heterogeneity -- but still some you can control for.

Second, one of the big arguments is that it's not so much infections per se driving down consumption, but fear about transmission that depresses expectations and consumer confidence. What is your answer to that? One way is to compare states that are similar in political affiliation, but different in state policies. But, that goes back to #1 in needing controls and placebos.

Third, the writing made it a little hard to read -- wordier in some places than it needed to be. There are also more citations you should make to recent work on consumption. Guerrieri et al have a paper on aggregate demand and supply from the pandemic; Yannelis and Pagela have a paper that came out at some point on consumption; etc.

I think you're on the right track, but it would be important to address these dimensions in your work going forward.

Reviewer #2: Comments to the Author

The authors explored spatial temporal patterns of COVID-19 incidence and NPIs and their relationship with indicators of economic activity in the US. Their results suggest that consumer spending patterns can be attributed to government mandates rather than COVID-19 incidence in the states.

This is a comprehensive and ambitious research project, representing a potentially useful and novel contribution to the literature. The paper is well written and interesting. However, I have some concerns about the methods used. My main concerns are about the choice of the indicators.

1. The authors should provide further background and clarification on the choice of the incidence of COVID-19 as the main indicator of the spread of COVID-19. It is well known that States/Cities consider multiple other indicators to systematically monitor the status eg https://www.cdc.gov/mmwr/volumes/69/wr/mm6934e3.htmhttps://forward.ny.gov/metrics-guide-reopening-new-york.

As a consequence, pros and cons versus other indicators considered by States/Cities to change NPIs policies (eg, test positivity rate, hospital admissions vs capacity, viral transmissibility) should be discussed within the limitations of this research, included in a sensitivity analysis and/or mentioned for readers awareness and future research.

2. I would invite the authors to reflect on the factors that explain a higher incidence of COVID-19 (eg, socio-demographic composition, access to testing, public transportation, variants) and if / how those can affect the relationship with the rest of the indicators being analyzed at State level.

3. Once the background on the validity of COVID-19 incidence as an indicator of spread has been further elaborated, I suggest the authors should justify the choice of the source of data and clarify how the incidence of COVID-19 is defined:

• What is the relationship of the New York Times dataset to national and state surveillance efforts CDC eg Geographic Differences in COVID-19 Cases, Deaths, and Incidence — United States, February 12–April 7, 2020 | MMWR (cdc.gov)?

• Is the definition of COVID-19 incidence exclusively related to new, lab confirmed cases, excluding probable cases? Are asymptomatic cases included?

• Is it important for this exercise to consider geographical differences in case detection and reporting?

4. I have similar comments as above for the choice and definition of card spending as a proxy for consumer spending. To further justify its validity as a measure of spending during the pandemic, the authors should clarify:

• What type of consumer expenses are pooled in that indicator (travel, transportation, food, entertainment, real estate, health, etc), in what composition.

• Why a pool of different types of expenses is the right proxy for the aims of this research; did the authors consider sensitivity analyses to a more restricted group - is the food analyses presented in Supplements restricted to essential purchases in supermarket, excluding restaurants and bars?

• Have the authors considered a sensitivity analyses focused on personal savings rates instead?

5. With regards to the analyses of NPIs, I would advise the authors to also refer to the fact that there are multiple combinations of NPIs measures that are associated with different effectiveness. This element might have affected States differently, too, on top of overall NPIs stringency and compliance.

6. Potential suggestion to the authors: it may not be within the scope of this research, but I was curious if you conducted any exploratory quantitative threshold analyses across NPIs levels or phases eg what % change (range) in consumer spending was observed during the lockdown (vs pre-lockdown), and so on during phases of moderate and complete NPIs relaxations.

6. PLOS authors have the option to publish the peer review history of their article (what does this mean?). If published, this will include your full peer review and any attached files.

Reviewer #1: No

Reviewer #2: No

---

## [Author Response · Author response to Decision Letter 0]

4 Sep 2021

We thank the Editor and Reviewers for the insightful comments, which certainly guided us to improve the manuscript. We have done our best to address all the suggestions and questions. All responses are included in the file named "Response to Reviewers."

---

## [Decision Letter · Decision Letter 1]

10 Sep 2021

PONE-D-21-21306R1Relationship among state reopening policies, health outcomes and economic recovery through first wave of the COVID-19 pandemic in the U.S.PLOS ONE

Dear Dr. Ligo,

Thank you for submitting your manuscript to PLOS ONE. After careful consideration, we feel that it has merit but does not fully meet PLOS ONE’s publication criteria as it currently stands. Therefore, we invite you to submit a revised version of the manuscript that addresses the points raised during the review process.

We look forward to receiving your revised manuscript.

Kind regards,

Martial L Ndeffo Mbah, Ph.D

Academic Editor

PLOS ONE

Journal Requirements:

Additional Editor Comments (if provided):

Mainly address the first comment of reviewer 1.

Reviewers' comments:

Reviewer's Responses to Questions

**Comments to the Author**

1. If the authors have adequately addressed your comments raised in a previous round of review and you feel that this manuscript is now acceptable for publication, you may indicate that here to bypass the “Comments to the Author” section, enter your conflict of interest statement in the “Confidential to Editor” section, and submit your "Accept" recommendation.

Reviewer #1: All comments have been addressed

Reviewer #2: All comments have been addressed

2. Is the manuscript technically sound, and do the data support the conclusions?

Reviewer #1: Yes

Reviewer #2: Yes

3. Has the statistical analysis been performed appropriately and rigorously? 

Reviewer #1: Yes

Reviewer #2: Yes

4. Have the authors made all data underlying the findings in their manuscript fully available?

Reviewer #1: Yes

Reviewer #2: Yes

5. Is the manuscript presented in an intelligible fashion and written in standard English?

Reviewer #1: Yes

Reviewer #2: Yes

6. Review Comments to the Author

Reviewer #1: Thank you authors. I think this is a good improvement. I am on board with the econometric approach and results, but want to flag two important areas for this to be more presentable.

First, substantively: are you controlling for time and state fixed effects in your DD regression? I could not tell by the table formulation and your discussion of them. This is important. Put the temporal fixed effects in at least with the month and year aggregation. You need to isolate variation within the same state over time. To further drive home your point, you could always do a mediation analysis (R has a nice package) where you look at how the unemployment rate may serve as a mediating factor. The main point here is to ensure you're doing the DD right since those coefficients look a little big.

Second, presentation: the plots and tables don't look as good as they could. I know it might sound trite, but just take a look at tables in top economics journals, like QJE/AER. You want to make clear the coefficients of interest, have table notes that describe what you're doing (and the controls that are included), the sample period, etc. The tables you have for the DD are not at all conventional -- basically the best thing to do is an event study where you're plotting the coefficients on the post period 1 month after the removal of a lockdown all the way up to however many months you care about (maybe 6), with some before months too. You also want to put confidence intervals on them and ensure you're clustering at the state level.

(note that if you're putting in state FE etc, make sure you have all 50 states -- clustering with less than 40 obs [and here it's the states that matter] will give biased estimates)

Reviewer #2: I appreciate the comprehensive responses provided to all of my questions and comments. The resulting changes in the manuscript are clear and exhaustive. The paper appears in very good shape now and I recommend its acceptance for publication in PLOS ONE.

7. PLOS authors have the option to publish the peer review history of their article (what does this mean?). If published, this will include your full peer review and any attached files.

Reviewer #1: No

Reviewer #2: No

---

## [Author Response · Author response to Decision Letter 1]

21 Oct 2021

We thank the Editor and Reviewers for the insightful comments, which certainly guided us to improve the manuscript in this second revision. We have done our best to address all the suggestions and questions raised in the second review. The changes are summarized in the "Response to Reviewers" document.

---

## [Decision Letter · Decision Letter 2]

2 Nov 2021

Relationship among state reopening policies, health outcomes and economic recovery through first wave of the COVID-19 pandemic in the U.S.

PONE-D-21-21306R2

Dear Dr. Ligo,

We’re pleased to inform you that your manuscript has been judged scientifically suitable for publication and will be formally accepted for publication once it meets all outstanding technical requirements.

Kind regards,

Martial L Ndeffo Mbah, Ph.D

Academic Editor

PLOS ONE

Additional Editor Comments (optional):

Reviewers' comments:

Reviewer's Responses to Questions

**Comments to the Author**

1. If the authors have adequately addressed your comments raised in a previous round of review and you feel that this manuscript is now acceptable for publication, you may indicate that here to bypass the “Comments to the Author” section, enter your conflict of interest statement in the “Confidential to Editor” section, and submit your "Accept" recommendation.

Reviewer #1: All comments have been addressed

Reviewer #2: All comments have been addressed

2. Is the manuscript technically sound, and do the data support the conclusions?

Reviewer #1: Yes

Reviewer #2: Yes

3. Has the statistical analysis been performed appropriately and rigorously? 

Reviewer #1: Yes

Reviewer #2: Yes

4. Have the authors made all data underlying the findings in their manuscript fully available?

Reviewer #1: Yes

Reviewer #2: Yes

5. Is the manuscript presented in an intelligible fashion and written in standard English?

Reviewer #1: Yes

Reviewer #2: Yes

6. Review Comments to the Author

Reviewer #1: Thanks all for the good edits. I think the paper reads more clearly and the results are stronger. This topic has become so politicized that you just want the results to be as clean and justifiable as possible, so the added tables where you show heterogeneity for high income consumer spending and with the sequential controls included are all good. The one area you may want to think further about is how you're talking about recovering unbiased coefficients. I am not seeing the R-squared in the tables. If it is really that big (as you indicate in the text), that's very interesting and might explain the weaker significance in some specifications, but you should definitely replace the F stat in the tables with R square. F stat is not really important. The CI is also not needed - as long as you keep the SE and the stars, that is enough.

Reviewer #2: My prior comments were addressed. This is an important and timely contribution to a question of considerable importance. The analysis is overall well conducted and clearly reported. In my view, the manuscript can be accepted.

7. PLOS authors have the option to publish the peer review history of their article (what does this mean?). If published, this will include your full peer review and any attached files.

Reviewer #1: No

Reviewer #2: No

---

## [Editor Report · Acceptance letter]

4 Nov 2021

PONE-D-21-21306R2 

Relationship among state reopening policies, health outcomes and economic recovery through first wave of the COVID-19 pandemic in the U.S. 

Dear Dr. Ligo:

I'm pleased to inform you that your manuscript has been deemed suitable for publication in PLOS ONE. Congratulations! Your manuscript is now with our production department. 

Kind regards, 

on behalf of

Dr. Martial L Ndeffo Mbah 

Academic Editor

PLOS ONE